# Motivational Correlates, Satisfaction with Life, and Physical Activity in Older Adults: A Structural Equation Analysis

**DOI:** 10.3390/medicina59030599

**Published:** 2023-03-17

**Authors:** Filipe Rodrigues, Miguel Jacinto, Nuno Couto, Diogo Monteiro, António M. Monteiro, Pedro Forte, Raul Antunes

**Affiliations:** 1ESECS—Polytechnic of Leiria, 2411-901 Leiria, Portugal; 2Life Quality Research Center, 2040-413 Leiria, Portugal; 3Sports Science School of Rio Maior, Polytechnic Institute of Santarém, 2040-413 Rio Maior, Portugal; 4Research Centre in Sports Sciences, Health, and Human Development, 6201-001 Covilhã, Portugal; 5Department of Sport Sciences and Physical Education, Polytechnic of Bragança, 5300-253 Bragança, Portugal; 6Department of Sports, Higher Institute of Educational Sciences of the Douro, 4560-708 Penafiel, Portugal; 7CI-ISCE, ISCE Douro, 4560-708 Penafiel, Portugal; 8Center for Innovative Care and Health Technology (CiTechcare), 2410-541 Leiria, Portugal

**Keywords:** motivation, satisfaction with life, physical activity, structural equation modeling

## Abstract

*Background:* Motivation is a crucial factor in predicting health-related outcomes, and understanding the determinants of motivation can provide valuable insights into how to improve health behaviors and outcomes in older adults. In this study, we aimed to investigate the associations between intrinsic and extrinsic exercise motivation, basic psychological needs, satisfaction with life, and physical activity among the elderly population. *Methods*: The sample consisted of 268 older adults (59 male, 209 female) aged 65–90 years old (Mage = 68.11, SD = 6.95). All participants reported that they were exercising, on average, 1.65 days (SD = 0.51) per week. Factor analysis was conducted using a two-step approach. First, a confirmatory factor analysis and then a structural equation model considering all variables under analysis was performed. *Results*: the structural model displayed acceptable fit to the data: χ2/df = 3.093; CFI = 0.913; TLI = 0.908; SRMR = 0.071; RMSEA 0.079 [0.066, 0.092]. Significant direct effects were found as theoretically proposed, namely: (a) intrinsic motivation were positively and significantly associated with basic psychological need satisfaction (*p* < 0.001); (b) extrinsic motivation were negatively but not significantly associated with basic psychological needs (*p* < 0.001); and (c) basic psychological need satisfaction were positively and significantly associated with satisfaction with life (*p* < 0.001) and physical activity (*p* < 0.001). *Conclusions*: Intrinsic motivation and basic psychological needs play a crucial role in shaping exercise behavior and overall well-being. By understanding these motivation and needs, exercise and health professionals can work towards fulfilling them and achieving a greater sense of satisfaction in the life of the elderly and promote exercise adherence.

## 1. Introduction

Physical activity is an important part of healthy aging for older adults. Regular physical activity, also known as exercise, can help prevent or delay the onset of a variety of chronic diseases, including coronary heart disease, stroke, type 2 diabetes, and several cancers [1]. Exercise can also improve mental health, increase cognitive performance, and lower the risk of falls in older people [2]. Low-impact exercises such as walking, swimming, and resistance training are among the most recommend kinds of physical activity for older people [3]. These activities can help increase strength and flexibility, lowering the chance of falling and other accidents [4,5]. Regular physical activity can help improve overall strength, balance, and coordination, improve mood, and reduce stress and anxiety [6,7]. Participating in physical activity in a group setting can increase interaction and solidarity among older adults [8]. Thus, exercise is an essential part of maintaining good physical and mental health among elderly people. However, physical inactivity is a growing health concern among the elderly population, and it carries a significant amount of risk [9]. The lack of physical activity and high engagement in sedentary behaviors can accelerate the aging process and lead to poor mental health, and difficulty in performing everyday tasks [4].

According to the existing literature, exercise requires some sort of motivation to be done on a regular basis [10]. It is essential for developing and maintaining habits throughout time. Without motivation, finding the energy and enthusiasm to exercise or be physically active becomes difficult. In this regard, motivation has a two-fold effect on exercise. On the one hand, the proper form of motivation aids in the adoption of physically active behaviors [11,12] and promotes people to continue pushing themselves to achieve their goals in the long term. A lack of motivation, on the other hand, might lead to a lack of exercise frequency or even dropout [13]. Therefore, reasons for engaging in regular exercise could provide evidence for the creation and promotion of regular exercise among the elderly [14].

### 1.1. Motives for Exercise Practice

Motivation is a significant driver for older adults to maintain a healthy lifestyle and to stay active [15,16]. It has been proven to have a positive effect on overall mental and physical health, as well as improving cognitive functioning and memory [17]. Within the realm of motivational factors, motivation explain “what” a person expects to achieve from engaging in exercise (e.g., “I exercise to improve my competence”). Motivation, on the other hand, is concerned with the “why” of the individual committing to exercise (e.g., “I exercise because I want to be recognized by my friends as someone healthy”).

Ryan and Deci [10] created the goal-content theory to comprehend how motivation and reasons might result in diverse outcomes that affect behavior performance. This theory is a psychological theory that proposes that behavior is motivated by the goals a person intends to achieve. According to the theory, motivation-directed behavior is a result of external or internal factors. Intrinsic motivation originates from inside and is based on personal interests and aspirations. This form of motivation is especially crucial for older adults, since it can help them stay focused and involved while also delivering a stronger feeling of purpose and satisfaction with life [18,19]. Extrinsic motivation is when older adults are driven to do something for an external reward, such as a recognition from others, competition, or affiliation [20]. Common extrinsic motivations for exercising include wanting to look good, to feel better, or to receive praise from friends and family [10].

Individuals are motivated by a wide range of motivation and goals, but, at their core, most of these goals are rooted in satisfying basic psychological needs [10]. These needs can provide a sense of purpose and direction and striving to meet them can bring a sense of satisfaction. Understanding these needs can help a person stay focused and motivated to reach their goal [21].

### 1.2. Basic Psychological Needs

The basic psychological needs theory proposed by Ryan and Deci [10] describes that humans have three fundamental needs that must be fulfilled for them to be motivated, have a sense of purpose, and live a healthy and meaningful life. The need for autonomy is the need for individuals to be able to make their own choices and decisions without feeling controlled by external forces. The need for competence is the need to feel capable and competent in the tasks they take on. The need for relatedness is the need to form meaningful connections with other people and to feel accepted as part of a larger social context [22].

This theory emphasizes that these three needs must be met for a person to experience growth, development, and commitment to exercise [23]. It also suggests that when these needs are not met, individuals may experience unhealthy coping mechanisms such as procrastination, avoidance, and/or exercise dropout [24]. Therefore, it is important for individuals to focus on actively engaging in activities that fulfill their needs to lead a meaningful life, especially in the elderly, since regular physical activity may increase positive judgements regarding their affective state [25].

### 1.3. Associations between Motivational Factors, Satisfaction with Life and Physical Activity

Satisfaction with life, according to the works of Diener [26,27,28], is the extent to which an individual experiences positive emotions, such as joy, contentment, and pride, and a general feeling of well-being and life satisfaction. This well-being indicator looks at how satisfied individuals feel with their personal accomplishments, relationships, environment, and other aspects of life. Psychological needs, such as the need to belong to a social group, the desire for self-actualization, and the desire for personal commitment, are necessities for life and growth. Research has found that when basic psychological needs are met, it has a positive effect on an individual’s overall satisfaction with life [29]. Satisfying these needs leads to a sense of security, contentment, and purpose, which leads to higher satisfaction with life. Conversely, when these needs are not met, it can lead to feelings of insecurity and unease, which will negatively affect satisfaction with life [10].

The connection between basic psychological needs and exercise are strong and indisputable. Studies have found that when these needs are met, individuals are more likely to engage in regular exercise [30]. This is because when individuals feel connected to a larger environment, have a sense of purpose, and feel competent, they are more likely to seek out active behaviors [10]. Therefore, it is important to recognize and address the psychological needs of older adults to encourage exercise and overall health and associated satisfaction with life.

### 1.4. Current Research

Individuals may have some difficulties judging satisfaction with their entire life (across domains) and consequently subjectively partition their experiences into specialized categories. In response to the drawbacks of global measures, previous research [19,31] has advocated investigating in individual domains of life satisfaction that are specific to context and are relevant to other populations beyond adolescents and adults. We know little about whether exercise motivation and basic psychological needs may be responsible for generating higher well-being outcomes, because most previous studies have examined satisfaction with one’s life as a whole, as described by Nakamura [28]. The existing research has been foundational and has made significant contributions to the literature [30,31,32], but several gaps limit its capacity to demonstrate how motivational factors relate to well-being in the elderly population.

This study intends to contribute to the existing literature by recognizing the uniqueness of exercise in the older adult population, considering the reasons for engaging in regular physical activity. Although older adults are assumed to engage in regular physical activity for self-determined reasons and motivation [30,33], we can presume that being active is essential to them, which could explain why they have more satisfaction with life compared to physical inactive older adults. The current study adds to prior research on motivation to exercise [34,35,36] by studying how motivation and basic psychological needs could be connected with measures of well-being such as life satisfaction, and also the association with physical activity. Thus, depending on the motivation and satisfaction of basic psychological needs experienced during exercise, the findings could provide direction on how to manipulate physical activity as a means of promoting life satisfaction and exercise adherence. In this study, we aimed to investigate the associations between intrinsic and extrinsic exercise motivation, basic psychological needs, satisfaction with life, and physical activity among the elderly population using structural equation modeling procedures. According to the above-mentioned literature, we hypothesize that: (a) intrinsic motivation would be positively and significantly associated with basic psychological needs, satisfaction with life, and physical activity; (b) extrinsic motivation would be negatively and significantly associated with basic psychological needs, satisfaction with life, and physical activity; (c) autonomy, competence, and relatedness would be positively and significantly associated with satisfaction with life, and physical activity.

## 2. Methods

### 2.1. Participants

This study had a cross-sectional design and the sample consisted of 268 older adults (59 male, 209 female) aged 65–90 years old (Mage = 68.11, SD = 6.95). All participants reported that they were exercising, on average, 1.65 days (SD = 0.51) per week. For inclusion, those who met the following inclusion criteria were considered: (i) aged 65 years old or older; (ii) provide informed consent to participate; and (iii) be physically active, engaging at least 1 day per week on structured and organized exercise activity.

### 2.2. Procedures

Ethical institutional approval (128/CES/INV/2013) was obtained prior to conducting this study. Following ethical institutional approval, senior universities were contacted (*n* = 3). A convenience sampling method was used for data collection, because the researchers could have access to potential participants. Objectives and data collection procedures were explained individually to the principles and managers. After obtaining approval (*n* = 3), potential participants were contacted during class and asked to participate voluntarily in this study. Objectives for this study were explained to all participants, and signed informed consents were obtained individually. Participants completed measures using a paper-and-pencil form questionnaire. The mean time to complete the questionnaires was less than 15 min. Older adults received no compensation for participating in this study but were thanked for their contribution.

### 2.3. Instruments

The Goal-Content Exercise Questionnaire Portuguese version for the older adult population [37] was used for measuring intrinsic and extrinsic goals that individuals can pursue in the exercise context. This measure comprises 17 items evaluated on a seven-point Likert scale ranging from 1 (“totally disagree”) to 7 (“totally agree”). Subsequently, the items were grouped into five factors: health management (e.g., “to improve my overall health”; skills development (e.g., “to acquire new exercise skills”); affiliation (e.g., “to develop close friendships”); image (e.g., “to improve my appearance”); and social recognition (e.g., “to be socially respected by others”). Each factor is the mean score of three or four items.

Participants completed the Basic Psychological Need Satisfaction Portuguese version for the elderly [24]. This scale assesses perceived autonomy, competence, and relatedness satisfaction. It is multidimensional, and the scale is composed of 21 items, seven for each construct. The participants indicated their agreement to each item through a seven-point Likert-type scale with response choices that varied between 1 (“totally disagree”) and 7 (“totally agree”).

Participants completed the Satisfaction with Life Scale Portuguese version for the elderly [38]. This five-item scale is designed to measure cognitive judgements of one’s life satisfaction and study participants responded to each item using a seven-point scale ranging from 1 (“totally disagree”) to 7 (“strongly agree”).

Participants were asked once to report their weekly exercise frequency over the last week (statement: “How many days per week do you think you have exercised over the last week?”). For clarity, examples of exercise behavior were provided in the questionnaire [39].

### 2.4. Statistical Analysis

Using IBM SPSS STATISTICS 25.0 version (IBM Corp., Armonk, NY, USA) software, descriptive statistics such as means and standard deviations, as well as bivariate correlations between all variables under consideration, were generated. To calculate the statistical significance of a deviation from the normal distribution, the skewness and kurtosis estimations were divided by their standard errors to yield the z-score. A z-score less than |1.96| indicated a normal distribution. For the referred analyses, a significance value ≤ 0.05 was assumed to reject the null hypothesis [40]. Internal consistency coefficients were calculated considering coefficients above 0.70 as acceptable [41].

Afterwards, following Kline’s [42] recommendations, factor analyses were conducted using a two-step approach. First, a confirmatory factor analysis and then a structural equation model considering all variables under analysis was performed. These analyzes were carried out in Mplus 7.4 [43]. Full information robust maximum likelihood was used to deal with the small amount of missing data at the item level (random missing = 3%), as proposed by Savalei [44]. For model fit, several traditional and incremental indexes were considered, namely Comparative Fit Index (CFI), Tucker–Lewis Index (TLI), Root Mean Square Error of Approximation (RMSEA) and its respective 90% Confidence Interval (CI90%). Scores of CFI and TLI ≥ 0.90, and RMSEA ≤ 0.08 were indicative of acceptable fit, as proposed by several authors [45,46]. The chi-square test, χ^2^, and the degrees of freedom will be reported for visualization purposes but not examined, as they are both affected by the complexity of the model and sample size [46]. Direct and indirect effects were analyzed according to standardized coefficients and their respective 95% Confidence Interval (CI95%). Regression paths were considered significant if the CI95% did not include zero [47].

## 3. Results

Table 1 displays descriptive statistics as well as bivariate correlations and internal consistency coefficients. The mean scores for health motivation were higher than those for the other exercise motivation. Z-scores were both less than the cut-off for suggesting normal distribution. As expected, several significant bivariate correlations emerged, namely: (a) intrinsic motivation were positively and significantly associated with autonomy, competence, relatedness, satisfaction with life (except skill development), and physical activity (*p* < 0.01); and (b) autonomy, competence, and relatedness were positively and significantly associated with satisfaction with life (*p* < 0.01), and physical activity (*p* < 0.01). Interestingly, extrinsic motivation was positively and significantly associated with all basic psychological needs (*p* < 0.01) and satisfaction with life (*p* < 0.01). Alpha coefficients were above adequate, showing acceptable internal consistency (>0.70).

Due to model complexity, composite factors for intrinsic (health and skill development) and extrinsic (affiliation, image, and social recognition) motivation, as well as basic psychological need satisfaction (autonomy, competence, and relatedness) were created. Since the first-order factors displayed acceptable reliability coefficients, and the composite factors also (intrinsic motivation α = 0.70; extrinsic motivation α = 0.81; basic psychological need satisfaction α = 0.73), we tested a more parsimonious model (see Figure 1). The measurement model provided acceptable fit to the data: χ^2^/df = 3.084; CFI = 0.924; TLI = 0.918; SRMR = 0.067; RMSEA 0.078 [0.065, 0.102]. Considering these results and the acceptable internal consistency coefficients, we moved forward on testing the structural model. Looking at the results, the structural model displayed acceptable fit to the data: χ^2^/df = 3.093; CFI = 0.913; TLI = 0.908; SRMR = 0.071; RMSEA 0.079 [0.066, 0.092].

Direct regression paths were analyzed (results are displayed in Figure 1). Overall, significant direct effects were found as theoretically proposed, namely: (a) intrinsic motivation was positively and significantly associated with basic psychological need satisfaction (*p* < 0.001); (b) extrinsic motivation was negatively but not significantly associated with basic psychological needs (*p* < 0.001); and (c) basic psychological need satisfaction was positively and significantly associated with satisfaction with life (*p* < 0.001) and physical activity (*p* < 0.001). Table 2 shows indirect effects between variables. Regarding indirect effects, a significant result emerged in the relationship between intrinsic motivation and satisfaction with life. There is preliminary evidence that basic psychological need satisfaction shows a mediating role in the relationship between intrinsic motivation and satisfaction with life. Results will be discussed accordingly.

## 4. Discussion

We aimed to investigate the associations between intrinsic and extrinsic exercise motivation, basic psychological needs, satisfaction with life, and physical activity among the elderly population. According to the above-mentioned literature, current results from the structural equation model analysis partially support the proposed hypotheses, namely: (a) intrinsic motivation was positively and significantly associated with basic psychological needs, satisfaction with life, and physical activity; (b) extrinsic motivation was not associated with basic psychological needs, satisfaction with life, and physical activity; (c) basic psychological needs were positively and significantly associated with satisfaction with life and physical activity.

The results showed that intrinsic motivation was positively and significantly associated with basic psychological needs as theoretically expected [10]. This result supports previous empirical research [48] showing that health and skill development are significantly related to the satisfaction of autonomy, competence, and relatedness. Since intrinsic motivation refers to the internal forces that drive individual behavior, higher levels of intrinsic reasons lead to a sense of personal accomplishment, a desire for self-expression, and the need for social connection. Specifically, based on bivariate correlation results, both intrinsic motivations were significantly correlated with all three basic psychological needs. Intrinsically motivated elderly people seem to have a sense of autonomy, since they choose to exercise based on motivation, which they consider more significant [25]. Thus, a sense of achievement arises from having more autonomy for self-regulation, choice, and volition in exercising. In addition, the results support the theory by showing that when the elderly are given opportunities to develop their skills and abilities, they tend to be more satisfied with their lives, and feel more motivated to achieve their goals. Having a sense of competence that comes from achieving an intrinsic goal or mastering a new skill can provide a sense of self-worth, as individuals who feel they are capable tend to feel more positive about themselves [24]. Results seem to indicate that older adults who have positive relationships with others tend to be more satisfied with their lives, since their goals are intrinsically driven. The sense of belonging that comes from being part of a group or community can be experienced based on a sense of self-improvement [49].

Extrinsic motivation was not significantly associated with basic psychological need satisfaction, as seen in Figure 1. Research has shown that extrinsic motivation can be effective in making individuals engage in certain behaviors, but it may not always lead to long-term motivation or overall well-being [10,19]. The bivariate results indicated a significant association between affiliation, image, and social recognition, but the significance of the associations were nullified in the structural equation model, indicating that intrinsic motivation has a significant association in explaining the satisfaction of basic psychological needs. Thus, while extrinsic motivation has a relationship with autonomy, competence, and relatedness, these associations may vanish in the long term [23]. For example, when an elderly person is extrinsically motivated to complete an exercise, they may do so to receive praise or social recognition, but they may not find the task enjoyable or satisfying. However, this is speculative, and more empirical research is needed.

Basic psychological needs displayed a significant association with satisfaction with life and exercise frequency. Greater experience of autonomy, competence, and relatedness satisfaction leads to a greater sense of self-worth and acceptance with one’s life. Supporting previous research [10,19,24] when these basic psychological needs are met, individuals tend to be more self-motivated and self-directed, which can lead to greater well-being and psychological adjustment. For example, when the elderly are given opportunities to make their own choices on the exercises they like, develop their skills and abilities, and have positive relationships with others, they tend to be more satisfied with their lives and feel more motivated to achieve their goals. The same trend appears in the relationship between basic psychological needs and exercise frequency. Basic psychological need satisfaction and exercise are closely associated, with basic psychological needs relating to the innate wants for autonomy, competence, and relatedness, and exercise being an activity that enhances general psychological health [10]. The need for autonomy is a key factor in motivating physical activity. This need allows individuals to have control over their lives and activities, which has been shown to positively influence exercise adherence [13]. Additionally, having a sense of competence can lead individuals to engage in more regular activity. Lastly, having a sense of relatedness, or a feeling of connection to the environment and peers, has been associated with increased levels of physical activity in the elderly population [21,24].

According to the present results, the need for autonomy, competence, and relatedness seems to display a mediating role in the relationship between intrinsic motivation and satisfaction with life (see Table 2). The mediating role of basic psychological needs refers to the way in which these innate needs act as a link between personal factors (i.e., exercise motivation) and outcomes (i.e., satisfaction with life). According to research, when basic psychological needs are addressed, individuals become more self-motivated and self-directed, which can lead to improved well-being and psychological adjustment, based on a positive environment or intrinsically driven motivation [10]. Current results support this assumption, since elderly people who feel in control of their own decisions regarding the exercise program and are able to make choices that align with their values, and they tend to have higher levels of well-being and life satisfaction. Additionally, when older adults feel competent and effective in exercising, they tend to have higher well-being. Finally, when these individuals feel connected and attached to others, also known as a sense of “tribe”, the elderly tend to have higher levels of social support, which can also promote well-being. This is driven by health and skill development motivation, as they are associated with basic psychological needs. Thus, the satisfaction of basic psychological needs appears to mediate the relationship between intrinsic motivation and well-being, meaning that the delight and pleasure obtained from engaging in intrinsically motivated activities is contingent on the satisfaction of basic psychological needs.

The current study contributes to our understanding of the cognitive mechanisms that underpin both exercise frequency and life satisfaction in a sample of older adults. Our findings support the hypothesis that intrinsic motivation can increase well-being indicators such as satisfaction with life, based on the satisfaction of three innate basic psychological needs that elderly people have throughout an exercise session. To the extent that long-term exercise adherence seems to be driven by self-determined motivation, exercise professionals may modify their interpersonal behaviors and communication styles to achieve adaptive outcomes at the contextual (e.g., energized towards exercise activities) and general (e.g., overall well-being and satisfaction with life) levels.

### Limitations, Strength, and Agenda for Future Research

Although this cross-sectional study has many advantages, it also has several limitations that should be considered when interpreting the results. Participants were regular exercisers recruited from senior universities that provided regular exercise activities. Because this study was conducted in Portugal, the findings cannot be generalized to other cultures and circumstances. Future research should design longitudinal or prospective methods to test causal relationships as well as consider other sample characteristics to discuss variability among studies (e.g., health and fitness center, community programs). Another limitation is the consideration of only adaptive outcomes. Forthcoming studies should consider mal-adaptive outcomes (e.g., anxiety) and exercise dropout that could arise from low need satisfaction or even need frustration.

This study focused on strong statistical testing and theoretical models in an under-researched context of older adults and should be considered to be one of its strengths. As a result, this research has produced evidence of the cognitive response to exercise and its relationship to life satisfaction. The current study offers a preliminary look at the mechanisms that may translate intrinsic motivation into a more positive and subjective experience of exercise, ultimately translating into well-being and self-esteem by satisfying three basic psychological needs. To better understand the mechanisms underlying this relationship, the replication and extension of current findings using additional potential mediators such as behavioral regulation, affective response, and objectively measured outcomes (e.g., exercise participation using attendance records) is required.

## 5. Conclusions

Intrinsic motivation and basic psychological needs are associated with exercise behavior and overall life satisfaction. By understanding these motivation and needs, exercise and health professionals can work towards fulfilling them and achieving a greater sense of satisfaction in the life of the elderly and promote exercise adherence. Basic psychological needs and life satisfaction are critical notions that shed light on the internal and environmental aspects that determine general well-being. The elderly can strive towards better fulfillment in life by knowing which exercise motivation promote autonomy, competence, and relatedness more significantly. As a result, it is critical for professionals and organizations to develop ways to address older adults’ basic psychological requirements while also supporting their general well-being.

## Figures and Tables

**Figure 1 medicina-59-00599-f001:**
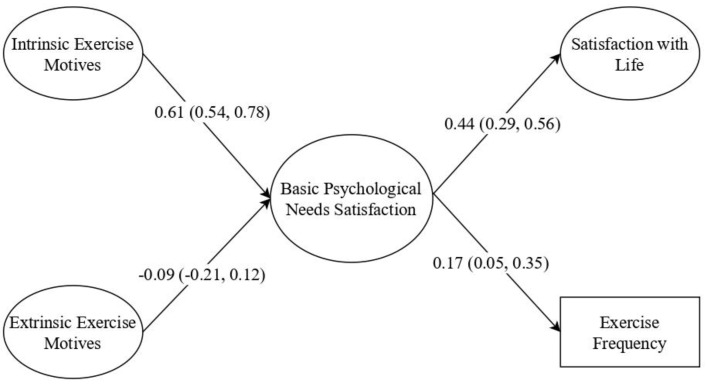
Structural equation model. **Notes:** Standardized coefficients are reported; within brackets = 95% confidence interval.

**Table 1 medicina-59-00599-t001:** Descriptive statistics, composite reliability coefficients, and latent correlations.

Variables	M	SD	S	K	1	2	3	4	5	6	7	8	9	10	α
1. Health Management	6.01	0.81	−0.85	1.44	1										0.83
2. Skill Development	5.03	1.18	−0.84	1.66	0.46 **	1									0.78
3. Affiliation	4.89	1.19	−0.90	1.59	0.37 **	0.64 **	1								0.73
4. Image	4.92	1.19	−0.80	1.42	0.44 **	0.47 **	0.50 **	1							0.81
5. Social recognition	3.81	1.44	−0.18	−0.43	0.11	0.50 **	0.65 **	0.60 **	1						0.71
6. Autonomy	5.55	0.97	−0.24	−0.01	0.38 **	0.28 **	0.15 *	0.18 **	0.07	1					0.72
7. Competence	5.13	0.87	−0.44	1.47	0.32 **	0.31 **	0.32 **	0.27 **	0.28 **	0.39 **	1				0.76
8. Relatedness	5.61	0.74	−0.09	0.11	0.46 **	0.26 **	0.30 **	0.29 **	0.22 **	0.48 **	0.62 **	1			0.82
9. Satisfaction with Life	4.70	1.07	−0.41	0.48	0.18 **	0.02	0.01	0.19 **	0.09	0.35 **	0.32 **	0.31 **			0.86
10. Exercise Frequency	1.65	1.50	1.43	1.64	0.16 **	0.22 **	0.09	0.06	0.03	0.06	0.14	0.11	0.05	1	-

**Notes:** M = Mean; SD = Standard-Deviation; S = Skewness; K = Kurtosis; α = Internal consistency coefficient; * *p* < 0.05; ** *p* < 0.01.

**Table 2 medicina-59-00599-t002:** Indirect regression paths.

Regression Path	β	SE	*p*-Value
Intrinsic Motives → Basic Psychological Needs Satisfaction → Satisfaction with Life	0.27	0.21	<0.05
Intrinsic Motives → Basic Psychological Needs Satisfaction → Exercise Frequency	0.10	0.11	>0.05
Extrinsic Motives → Basic Psychological Needs Satisfaction → Satisfaction with Life	−0.4	0.19	>0.05
Extrinsic Motives → Basic Psychological Needs Satisfaction → Exercise Frequency	−0.01	0.09	>0.05

**Notes:** β = standardized regression path; SE = Standardized Error; *p*-value = significance level.

## Data Availability

Data are available under request to the first author.

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
