# Peer review of "Motivational Correlates, Satisfaction with Life, and Physical Activity in Older Adults: A Structural Equation Analysis"

_medicina, 2023, doi:10.3390/medicina59030599_

Round 1
Reviewer 1 Report
The manuscript is relevant to the field and presented in a well-structured manner. A statistical approach is considerably accurate. Detailed comments:
There is a very long introduction to this study. Please consider moving some parts to Discussion.
Line 150-151 This study makes a paramount contribution by recognizing the uniqueness of exercise in the older adult population considering the reasons for engaging in regular physical activity.
I feel like this sentence does not match the Introduction.
Line 266-268- no p-values are given here. You are also inconsistent in reporting p-values in the abstract.
Figure 1. Structural equation model -can you improve the readability of this figure by enlarging elements and adding p-value ( or symbol *) to facilitate interpretation?
Author Response
Reviewer 1
The manuscript is relevant to the field and presented in a well-structured manner. A statistical approach is considerably accurate. Detailed comments:
Response: We appreciate your positive view. Point-by-point responses as provided and revisions are tracked in the manuscript using the track change option in MS Word.
There is a very long introduction to this study. Please consider moving some parts to Discussion.
Response: The introduction section was shortened.
Line 150-151 This study makes a paramount contribution by recognizing the uniqueness of exercise in the older adult population considering the reasons for engaging in regular physical activity. I feel like this sentence does not match the Introduction.
Response: Sentence was revised
Line 266-268- no p-values are given here. You are also inconsistent in reporting p-values in the abstract.
Response: Revised.
Figure 1. Structural equation model - can you improve the readability of this figure by enlarging elements and adding p-value ( or symbol *) to facilitate interpretation?
Response: Figure was revised. We presented the IC95%, thus, adding p-value would confuse readers. As described in the manuscript: Regression paths were considered significant if the CI95% did not include zero.
Reviewer 2 Report
This manuscript has investigated the association between intrinsic and extrinsic exercise motives, basic psychological needs, satisfaction with life, and physical activity among 268 older adults. The following are several points that the authors should address.
1. Given the cross-sectional design of this study, the authors should avoid any indications of causality throughout the manuscript. For example, “Motivational determinants”à” Motivational correlates” in the title.
2. This is perspective, but not a conclusion, of your study. Please rephrase. The same applies to the conclusion section in the manuscript.
3. Too much content in the background. It seems to be difficult for future readers to follow your story. Please clarify what is known and unknown about this topic briefly.
4. Is there any specific reason that this study focuses on older adults who were physically active? Please clarify this point in the introduction.
5. In the structural equation analysis, please clarify the role of “Basic Psychological Needs” in the results. A mediator? The same applies to the conclusion.
Minor comment
Please insert the correct reference (Couto et tal.) in line 202.
Author Response
Reviewer 2
This manuscript has investigated the association between intrinsic and extrinsic exercise motives, basic psychological needs, satisfaction with life, and physical activity among 268 older adults. The following are several points that the authors should address.
Response: We appreciate your positive view. Point-by-point responses as provided and revisions are tracked in the manuscript using the track change option in MS Word.
1. Given the cross-sectional design of this study, the authors should avoid any indications of causality throughout the manuscript. For example, “Motivational determinants” to “Motivational correlates” in the title.
Response: Title was revised.
2. This is perspective, but not a conclusion, of your study. Please rephrase. The same applies to the conclusion section in the manuscript.
Response: We revised the entire manuscript. We clarified in the manuscript that this is a cross-sectional study (see participant and limitation subheading).
3. Too much content in the background. It seems to be difficult for future readers to follow your story. Please clarify what is known and unknown about this topic briefly.
Response: The introduction section was shortened.
4. Is there any specific reason that this study focuses on older adults who were physically active? Please clarify this point in the introduction.
Response: We collected data from active elderly as they are more easily to recruit. As described in the method section: Ethical institutional approval (128/CES/INV/2013) was obtained prior to conducting this study. Following ethical institutional approval, senior universities were contacted. A convenience sampling method was used for data collection, because the researchers could have access to potential participants. In addition, it seems logic to measure exercise motives from physically active elderly.
5. In the structural equation analysis, please clarify the role of “Basic Psychological Needs” in the results. A mediator? The same applies to the conclusion.
Response: Basic psychological needs are described as mediators in the relationship between personal traits and contextual cues (see the works of Ryan and Deci, 2002; 2017). Thus, we inserted these needs as mediators (i.e., in the middle) in the model following previous theoretical and empirical evidence. Discussion is provided in lines 320-335. Conclusion section was revised.
Minor comment
Please insert the correct reference (Couto et al.) in line 202.
Response: Done.
Round 2
Reviewer 2 Report
Additional comments:
Overall, the manuscript has been reasonably revised. Here is my additional comments. The description of the results and current conclusion could be improved, particularly for the role of psychological needs as a mediator. For example, intrinsic motive was associated with “Satisfaction with Life” through psychological needs.
Minor comment
In line 367, “play” should be deleted?
Author Response
Reviewer 2
Additional comments:
Overall, the manuscript has been reasonably revised. Here is my additional comments.
Response: We appreciate your positive view. Point-by-point responses as provided
and revisions are tracked in the manuscript using the track change option in MS
Word.
The description of the results and current conclusion could be improved, particularly for
the role of psychological needs as a mediator. For example, intrinsic motive was
associated with “Satisfaction with Life” through psychological needs.
Response: Paragraph revised and added.
Minor comment:
In line 367, “play” should be deleted?
Response: Revised.